# Protocol for the Psychosis Immune Mechanism Stratified Medicine (PIMS) trial: a randomised double-blind placebo-controlled trial of single-dose tocilizumab in patients with psychosis

Éimear M Foley ,[1,2] Sian Lowri Griffiths,[3] Alexander Murray,[3] Jack Rogers,[3] Fabiana Corsi-Zuelli,[3,4,5] Hannah Hickinbotham,[6] Ella Warwick,[3] Martin Wilson,[3] Muzaffer Kaser,[6,7] Graham K Murray ,[6,7] Bill Deakin,[8] Deepak Jadon,[9] John Suckling,[6,7] Nicholas M Barnes,[5] Rachel Upthegrove,[3,10] Golam M Khandaker,[1,2,11,12] for the PIMS Collaboration

For numbered affiliations see end of article.

**Correspondence to**
Professor Golam M Khandaker;
golam.khandaker@bristol.ac.uk

## ABSTRACT

**Introduction** Evidence suggests a potentially causal role of interleukin 6 (IL-6), a pleiotropic cytokine that generally promotes inflammation, in the pathogenesis of psychosis. However, no interventional studies in patients with psychosis, stratified using inflammatory markers, have been conducted to assess the therapeutic potential of targeting IL-6 in psychosis and to elucidate potential mechanism of effect. Tocilizumab is a humanised monoclonal antibody targeting the IL-6 receptor to inhibit IL-6 signalling, licensed in the UK for treatment of rheumatoid arthritis. The primary objective of this study is to test whether IL-6 contributes to the pathogenesis of first episode psychosis and to examine potential mechanisms by which IL-6 affects psychotic symptoms. A secondary objective is to examine characteristics of inflammation-associated psychosis.

**Methods and analysis** A proof-of-concept study employing a randomised, parallel-group, double-blind, placebo-controlled design testing the effect of IL-6 inhibition on anhedonia in patients with psychosis. Approximately 60 participants with a diagnosis of schizophrenia and related psychotic disorders (ICD-10 codes F20, F22, F25, F28, F29) with evidence of low-grade inflammation (IL-6≥0.7 pg/mL) will receive either one intravenous infusion of tocilizumab (4.0 mg/kg; max 800 mg) or normal saline. Psychiatric measures and blood samples will be collected at baseline, 7, 14 and 28 days post infusion. Cognitive and neuroimaging data will be collected at baseline and 14 days post infusion. In addition, approximately 30 patients with psychosis without evidence of inflammation (IL-6<0.7 pg/mL) and 30 matched healthy controls will be recruited to complete identical baseline assessments to allow for comparison of the characteristic features of inflammation-associated psychosis.

**Ethics and dissemination** The study is sponsored by the University of Bristol and has been approved by the Cambridge East Research Ethics Committee (reference: 22/EE/0010; IRAS project ID: 301682). Study findings will be published in peer-review journals. Findings will also be disseminated by scientific presentation and other means.

**Trial registration number** ISRCTN23256704.

## STRENGTHS AND LIMITATIONS OF THIS STUDY

⇒ Adopting a randomised controlled trial design and patient selection based on elevated level of interleukin 6 (IL-6) (in addition to other criteria) will help examine the causal role of IL-6, and the therapeutic potential of targeting IL-6 pathway, in psychosis.

⇒ The use of target-specific intervention (anti-IL6R monoclonal antibody tocilizumab) will help assess the clinical relevance of IL-6 and related upstream and downstream inflammatory cytokines in psychosis.

⇒ The use of neuroimaging, cognitive tests and extensive peripheral blood biomarker exploration before and after tocilizumab treatment to assess potential mechanisms of effect.

⇒ One dose of tocilizumab is unlikely to be sufficient to test the efficacy of this drug as a potential treatment for psychosis, which is not the intention of this study.

⇒ Tocilizumab inhibits both IL-6 classical and trans signalling, and consequently the trial will not be able to distinguish between the effects of modulating these two signalling pathways specifically in psychosis.

## INTRODUCTION
### Scientific background and study rationale

The neuroimmune hypothesis of schizophrenia proposes that mild peripheral immune activation gives rise to an inflammatory response in the brain and neurobiological changes associated with psychotic illness.[1–4] Meta-analytic evidence is clear that circulating concentrations of interleukin 6

(IL-6) and other inflammatory proteins, such as C reactive protein (CRP), are increased in patients with psychosis, including treatment-naïve first episode psychosis (FEP),[5] compared with controls.[6–9] Prospective cohort studies show that these indices of mild immune activation precede the onset of symptoms.[10 11] Furthermore, genetic variants known to increase IL-6 concentrations are associated with genetic risk of schizophrenia.[12 13] These Mendelian randomisation studies eliminate the possibility that raised IL-6 concentrations are a consequence of environmental exposures associated with schizophrenia, such as obesity and smoking and instead suggest that IL-6 has a causal role in psychosis. Extending this approach using the UK Biobank population, we found that genetically predicted levels of IL-6 were associated with reduced grey matter primarily in the middle temporal gyrus, a region whose gene expression profile is enriched for IL-6 pathway proteins and for neuropsychiatric disorder ontologies.[14] Moreover, clinical studies report correlations between IL-6 levels and structural brain changes in individuals with schizophrenia,[15] with reduced grey matter volume being exaggerated in patients with psychosis and elevated inflammatory cytokines.[16] Though this causal evidence strongly implicates IL-6, only an intervention study in patients can test the causal hypothesis.

The neuroimmune hypothesis generally assumes that microglia, the brain's resident immune cells, are activated and pathogenic in schizophrenia. This is supported by traditional neuropathological studies and initial in vivo positron emission tomography (PET) imaging studies,[17–19] possibly reflecting impaired cellular control of inflammation or oxidative defence. Inflammatory damage may also account for evidence of oxidative stress from MRS glutathione studies.[20] However, whether microglia are the direct target of IL-6 is unclear and it is not certain that IL-6 can cross the blood–brain barrier and/or increase its permeability to circulating inflammatory cells, cytokines and chemokines.[2 21] Additionally, it is increasingly uncertain whether microglial inflammation, as traditionally understood, occurs in schizophrenia.[22 23] Recent meta-analyses of PET radioligand binding studies report decreased rather than increased radioligand binding to activated microglia.[24 25] This may account for the unexpected lack of therapeutic benefit of the antimicroglial antibiotic, minocycline, in recent large clinical trials.[26 27] Furthermore, large transcriptomic studies in postmortem brains report no change or reduction in microglial gene expression but increases in astrocytic expression.[23 28–32] It is increasingly understood that both peripheral immune responses and brain glial function are regulated by specialised T cells (Tregs), a subset of which reside in brain parenchyma.[33 34] A novel proposal is that Treg hypofunction accounts for mild peripheral immune disinhibition and dysregulated astroglial–microglial interaction, such that microglia are driven into a developmental, synapse-pruning phenotype while astroglia disrupt neurotransmitter function.[33 34] Importantly, there are bidirectional interactions between IL-6 and Treg function.[34] Crucially,

we will measure IL-6 in addition to cellular and molecular markers of immune function and investigate how they correlate with central markers and clinical state.

Previous attempts of testing the inflammatory hypothesis in therapeutic clinical trials have been attempted. However, little evidence of overall efficacy has been found.[35] These trials have generally tested broad spectrum agents, such as non-steroidal anti-inflammatory drugs, with no attempt to stratify patients according to evidence of inflammation. A trial using tocilizumab, a humanised monoclonal antibody (mAb) against the IL-6 receptor, currently licensed in the UK for treatment of rheumatoid arthritis (RA) and severe COVID-19, reported no improvements in any clinical measure in a small sample of 36 patients with established schizophrenia.[36] However, as mentioned previously, no stratification by inflammatory markers or any mechanistic immune measures was applied. Low-grade inflammation is associated with poor response to antipsychotic drugs,[37] but immunotherapy is unlikely to be relevant for all patients with psychosis. Meta-analysis suggests that evidence of immune activation, defined by elevated CRP levels, is present in a quarter to one-third of patients with schizophrenia.[38] A randomised controlled trial of infliximab, an antitumour necrosis factor alpha mAb, reported that antidepressant response was associated with higher CRP levels at baseline,[39] suggesting that patients with evidence of immune activation may be better candidates for immunotherapy trials. As far as we are aware, no previous clinical trial has selected patients with schizophrenia based on evidence of immune activation.

Selection of patients with particular symptom profiles and/or stage of illness may also be a useful strategy that needs to be employed in immunotherapy trials for schizophrenia. A wide variety of symptoms occur in schizophrenia such as hallucinations, delusions, anhedonia, cognitive dysfunction and affective symptoms and presentation of these symptoms differ from one individual to another. Some symptoms may be more related to inflammation than others. For instance, meta-analytic data suggests that elevated proinflammatory cytokines are associated with negative psychotic symptomatology.[40] Moreover, a recent study from the Avon Longitudinal Study of Parents and Children (ALSPAC) birth cohort reported that out of 20 positive and negative symptoms, CRP is particularly associated with anhedonia and auditory hallucinations.[41] Lastly, results from work, we have completed to date as part of the Medical Research Council (MRC)-funded larger Psychosis Immune Mechanism Stratified Medicine (PIMS) collaboration (MR/S037675/1), suggest that anhedonia may be a promising target in early phases of established psychotic disorder. Anhedonia and amotivation are strongly associated with poor functional outcomes in depression and schizophrenia, and present a formidable barrier to returning to work or building relationships.[42 43] Patients with psychotic disorders also present with cognitive deficits in a range of domains.[44] Available antipsychotic medications

have a limited effect on poor cognitive functioning in psychosis.[45] Illness stage may also be of relevance. Meta-analytic data has revealed no differences in IL-6 levels between stable, medicated patients with schizophrenia and controls, although compared with controls, IL-6 levels were similarly elevated in patients with FEP and those with acute relapse.[7] A separate meta-analysis found evidence of elevated blood cytokine levels in acutely and chronically ill patients with schizophrenia.[6] Focusing on particular inflammation-related symptoms, such as anhedonia, and/or illness stage may increase the chance of success for immunotherapy trials.

### Proposed study

The proposed study is a UK multisite (Birmingham, Bristol and Cambridge) proof-of-concept, randomised, parallel-group, double-blind, placebo-controlled trial.

### Study aims and hypotheses

The primary aim of this trial is to examine potential mechanisms by which IL-6 affects anhedonia, psychotic symptoms and cognition. Our primary hypothesis is that inhibition of IL-6 signalling with a single intravenous infusion of anti-IL6R mAb, tocilizumab, in individuals with psychosis and elevated IL-6 at baseline, will attenuate symptoms of anhedonia and amotivation in patients with psychosis, relative to placebo. This will provide further evidence for a potential causal role of inflammation in psychosis. Our secondary hypothesis is that reduction in peripheral inflammation after tocilizumab infusion in patients with psychosis and evidence of inflammation will be associated with central measures of oxidative stress and relevant resting state brain function.

We will also conduct deep immunophenotyping of peripheral blood mononuclear cell subsets (CD4$^+$, CD8$^+$, Tregs, natural killer and natural killer-T cells, monocytes, and B cells) to characterise their absolute number, frequency and function. The level of IL-6/STAT3 signalling inhibition within both innate and adaptive immune cells will also be examined using multicolour flow cytometry with an established optimised pSTAT3 phosflow assay. This will help identify the potential cellular impact of peripheral inflammation in psychosis, which is largely unknown. Functional assessment of IL-6/STAT3 signalling in immune cell subsets and their response to exogenous IL-6 stimulation will inform abnormal immune response in psychosis and allow measurement of response to tocilizumab at the cellular level.

A secondary objective is to carry out an observational study to examine clinical and biomarker differences and similarities between patients with psychotic disorder with and without evidence of inflammation and healthy controls (HCs). We hypothesise that individuals with psychotic disorder and evidence of inflammation, compared with those without evidence of inflammation and HCs, will have increased symptoms of anhedonia and amotivation, poorer cognitive functioning, and cellular and brain-based measures of immune dysfunction.

## METHODS

This protocol has been prepared in accordance with the Standard Protocol Items: Recommendations for Interventional Trials (SPIRIT) 2013 statement.[46] Please see online supplemental eTable 1 for the SPIRIT checklist. The planned start date for the PIMS trial was 1 November 2021; however, this was delayed due to the coronavirus (COVID-19) pandemic. We began recruiting at our site in Birmingham in November 2022, and we soon expect recruitment to begin at our Bristol and Cambridge sites. The planned end date is 31 May 2024.

### Patient and public involvement

The study protocol was prepared in collaboration with individuals with lived experience of mental illness who contributed to the development of participant information sheet, consent forms (online supplemental appendix I–III) and data collection procedures.

### Study design and sample

See figure 1 for an overview of study design. Individuals residing in Birmingham, Bristol or Cambridge in the UK will be recruited. Approximately 60 participants with FEP and evidence of inflammation (ie, IL-6≥0.7 pg/mL) will be randomised to receive either one intravenous infusion of tocilizumab (drug) or normal saline (placebo). For the secondary, observational study, we will compare baseline characteristics of the intervention cohort with approximately 30 participants with psychosis without evidence of inflammation (ie, IL-6<0.7 pg/mL), and approximately 30 HCs across Birmingham and Cambridge. Participants without evidence of inflammation and controls will not be randomised as they will not receive any intervention. Neuroimaging will only be undertaken by those without MRI contraindications who have given specific informed consent for MRI. Participants not eligible or not consenting for MRI will take part in all other aspects of the study.

### Intervention

Single intravenous infusion of tocilizumab (4.0 mg/kg; max 800 mg in total) or normal saline given to participants with psychosis and evidence of inflammation. Tocilizumab blocks both IL-6 classic and trans-signalling—the latter being responsible for most of the inflammatory effects of IL-6—providing broad inhibition of IL-6 signalling and a strong test of a casual role for IL-6 in psychosis.[47] Tocilizumab is the first-in-class, humanised mAb against the IL-6R, commercially available and licensed in the UK for treatment of RA. The approved dosage of tocilizumab for treatment of RA is 2, 4 or 8 mg/kg; max 800 mg in total. In RA, a single tocilizumab infusion has shown to improve clinical and laboratory measures within 48 hours, with most noticeable results in 1–2 weeks.[48 49] The follow-up schedule for our study is in keeping with this observation.

### Eligibility criteria

We will recruit participants aged 18–40 years. Patient participants must meet International Classification of

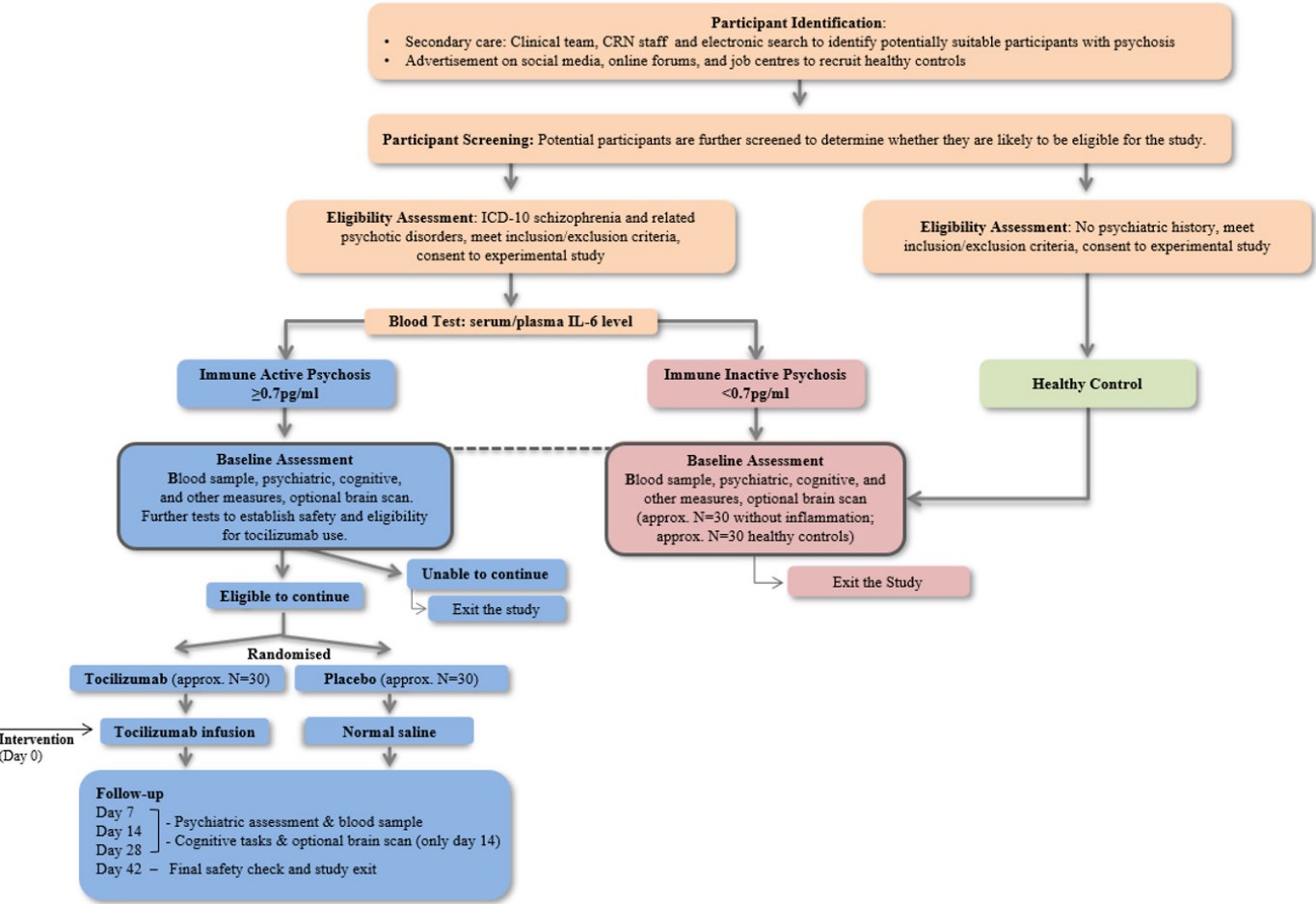

**Figure 1** Overview of study design. CRN, Clinical Research Network; ICD-10, International Classification of Diseases 10th Revision; IL-6, interleukin 6.

Diseases 10th Revision (ICD-10) criteria for a diagnosis of schizophrenia and related psychoses (ICD-10 code F20, F22, F25, F28, F29) at the time of eligibility assessment, be within 3 years of first diagnosis of psychotic disorder, be on a stable treatment regime with no recent (within 2 weeks) initiation, cessation or change in class of antipsychotic medication and have a Positive and Negative Syndrome Scale (PANSS) item score ≥3 on P1 (delusions), P2 (conceptual disorganisation), P3 (hallucinatory behaviour) or P6 (suspiciousness/persecution). Additionally, patients recruited to the interventional arm will be required to have serum IL-6 levels≥0.7 pg/mL and a Temporal Experience of Pleasure Scale (TEPS) anticipatory pleasure score ≤41 (based on item numbers 1, 3, 7, 11, 12, 14, 15, 16, 17 and 8) and consummatory pleasure score ≤36 (based on item numbers 2, 4, 5, 6, 8, 9, 10 and 13). The threshold of serum IL-6≥0.7 pg/mL as evidence of inflammation for this particular trial was chosen based on observations from the Personalised Prognostic Tools for Early Psychosis Management (PRONIA) cohort (https://www.pronia.eu). In 192 FEP patients included in the PRONIA study, the median value of serum IL-6 was 0.49 pg/mL (25th percentile 0.22 pg/mL; 75th percentile 1.11 pg/mL), and the mean was 0.79 pg/mL (SD±0.84). Based on these observations, we chose the cut-off of 0.7 pg/mL for patient selection in the current trial. Finally, COVID-19 anti-body titre test will be used to determine adequate levels of immune response via the following cut-offs (for poor response): 400 IU Roche/700 IU Abbot assay.

HCs will have no current or lifetime history of psychiatric diagnosis, as determined by the Mini-International Neuropsychiatric Interview (MINI). See table 1 for complete inclusion and exclusion criteria. HCs will be matched to patient participants at the group level by age and sex.

## Study outcomes

The primary outcome is anhedonia, defined as anticipatory and consummatory pleasure scores, assessed by the TEPS[50] at approximately day 14 post infusion. We will also collect data on several secondary/exploratory measures including (1) clinical outcomes, namely positive and negative symptoms of psychosis, depressive symptoms, fatigue, general quality of life and subjective well-being, (2) cognitive function (psychomotor speed, attention, memory and executive function), (3) neuroimaging outcomes based on comparisons of brain structure, function and oxidative stress levels using MRI and

**Table 1** Psychosis Immune Mechanism Stratified Medicine trial inclusion and exclusion criteria

| Group | Inclusion criteria | Exclusion criteria |
|---|---|---|
| All participants | ► Provide informed consent.<br>► Understand written and spoken English.<br>► Able and willing to consent to blood sampling.<br>► Willing to abstain from strenuous exercise for 72 hours prior to assessment. | ► Pregnancy (confirmed by urine pregnancy test) or breast feeding.<br>► Body mass index>35.<br>► Current or lifetime diagnosis of antisocial personality disorder, autism or other neurodevelopmental disorder, major traumatic brain injury.<br>► Currently active diagnosed eating disorder likely to compromise ability to take part.<br>► History of alcohol or substance use disorder (abuse/dependence) within 6 months prior to eligibility assessment (nicotine and caffeine dependence are not exclusionary).<br>► Current use of medication likely to compromise interpretation of immunological data.<br>► Known active current or history of recurrent bacterial, viral, fungal, mycobacterial or other opportunistic infections.<br>► Current infection with VZV, TB, Hepatitis B, Hepatitis C or HIV confirmed by blood test. X-ray of the chest will also be performed to assess for TB.<br>► Any major episode of infection requiring hospitalisation or treatment with intravenous antibiotics within 4 weeks of eligibility assessment.<br>► Unstable cardiac, pulmonary, renal, hepatic, endocrine, haematologic or active infectious disease, including current or prior malignancy.<br>► Diverticulitis, inflammatory bowel disease or uncontrolled gastric/duodenal ulcer.<br>► Concomitant autoimmune or autoinflammatory rheumatological disease.<br>► Concomitant treatment with any biologic drugs.<br>► Current and active ischaemic heart disease.<br>► Uncontrolled hypertension defined as systolic blood pressure>170 mm Hg or diastolic blood pressure>110 mm Hg.<br>► History of severe allergic or anaphylactic reactions to human, humanised or murine monoclonal antibodies.<br>► No history of chicken pox infection or no history of varicella zoster immunity. |
| Additional criteria for neuroimaging (optional) | ► Able and willing to consent to MRI scanning. | ► Contraindications to MRI. |
| Additional criterion for healthy controls | ► No current or lifetime psychiatric diagnosis. | |
| Additional criteria for all individuals with psychosis | ► Meet ICD-10 criteria for a diagnosis of schizophrenia and related psychoses (code F20, F22, F25, F28, F29) at the time of eligibility assessment, as determined by the Mini-International Neuropsychiatric Interview.<br>► Be within 3 years of first diagnosis of psychotic disorder.<br>► On stable treatment regime with no recent (within 2 weeks) initiation, cessation or change in class of antipsychotic medication.<br>► No indication or other reason for preclusion into research (eg, significant risk of suicidal behaviour or risk to others) as determined by their clinical team.<br>► Positive and Negative Syndrome Scale item score≥3 on P1 (delusions), P2 (conceptual disorganisation), P3 (hallucinatory behaviour) or P6 (suspiciousness/persecution). | |
| Additional criteria for intervention group | ► Serum IL-6 level≥0.7 pg/mL at eligibility and baseline assessment.<br>► Temporal Experience of Pleasure Scale anticipatory pleasure score≤41 (based on item numbers 1, 3, 7, 11, 12, 14, 15, 16, 17 and 8) and consummatory pleasure score≤36 (based on item numbers 2, 4, 5, 6, 8, 9, 10 and 13).<br>► Evidence of COVID-19 immunity required prior to infusion, confirmed before randomisation using evidence of vaccination and antibody titre test. | |
| Additional criterion for patients with psychosis without inflammation | ► Serum IL-6 level<0.7 pg/mL at eligibility and baseline assessment. | |

ICD-10, International Classification of Diseases 10th Revision; IL-6, interleukin 6; TB, tuberculosis; VZV, varicella zoster virus.

MRS outcomes, (4) blood biomarker outcomes, namely peripheral blood inflammatory markers, biochemical assays, including cortisol and cardiometabolic markers, and peripheral blood cellular immunophenotyping and (5) genetic outcomes including DNA and RNA sequencing and epigenetic mechanism assessment with methylation assays (see table 2 for more details). Where possible, blood samples will be collected during working hours and time of sampling will be recorded. However, a specified time window will not be given to ease burden on patients and to maximise participation.

**Table 2** Psychosis Immune Mechanism Stratified Medicine trial study measures

| Domain | Tool | Source | Validated tool | Time of assessment |
|---|---|---|---|---|
| Sociodemographic/lifestyle | Screening Questionnaire | Self-report | | Screening |
| | Medical History Questionnaire | Self-report/general practice | | Eligibility |
| | Substance Use Questionnaire | Self-report | | Eligibility |
| | Physical Measurements Form | Self-report | | Baseline |
| | Sociodemographic Questionnaire | Self-report | | Baseline |
| Psychiatric | The Temporal Experience of Pleasure Scale | Self-report | ✓ | Eligibility, baseline, follow-ups |
| | The Positive and Negative Syndrome Scale | Interviewer assessed | ✓ | Eligibility, baseline, follow-ups |
| | The Mini-International Neuropsychiatric Interview | Interviewer assessed | ✓ | Eligibility |
| | Psychiatric History Questionnaire | Self-report | | Baseline |
| | The Scale for the Assessment of Negative Symptoms | Self-report | ✓ | Baseline, follow-ups |
| | The Calgary Depression Scale for Schizophrenia | Interviewer assessed | ✓ | Baseline, follow-ups |
| | Multidimensional Fatigue Inventory | Self-report | ✓ | Baseline, follow-ups |
| | European Quality of Life-5 Dimensions Three-Level Version | Self-report | ✓ | Baseline, follow-ups |
| | Visual Analogue Scale for subjective well-being | Self-report | ✓ | Baseline, follow-ups |
| Cognitive | National Adult Reading Test for estimated premorbid IQ | Interviewer assessed | ✓ | Baseline, follow-up 2 |
| | CANTAB Reaction Time Test | Computer task | ✓ | Baseline, follow-up 2 |
| | Symbol Coding Test | Paper task | ✓ | Baseline, follow-up 2 |
| | CANTAB Rapid Visual Information Processing Test | Computer task | ✓ | Baseline, follow-up 2 |
| | CANTAB Paired Associates Learning Test | Computer task | ✓ | Baseline, follow-up 2 |
| | CANTAB One Touch Stockings of Cambridge Test | Computer task | ✓ | Baseline, follow-up 2 |
| Biologic | Inflammatory markers, cardiometabolic markers, IDO activation, white cell phenotyping | Laboratory tests | | Baseline, follow-ups |
| Genetic | RNA and DNA sequencing, methylation assay | Blood (RNA, DNA) | | Baseline, follow-ups |
| Neuroimaging | MRI Screening Questionnaire | | | Baseline, follow-up 2 |
| | Structural MRI, 1H-MRS measure of glutathione in the prefrontal cortex area, resting state fMRI | | | Baseline, follow-up 2 |

CANTAB, Cambridge Neuropsychological Test Automated Battery; 1H-MRS, proton magnetic resonance spectroscopy; IDO, indoleamine 2,3-dioxygenase; IQ, intelligence quotient.

## Sample size and statistical power

We will recruit approximately 60 patients with psychosis. However, currently there are no trials of immunotherapies for anhedonia in schizophrenia making accurate power calculation difficult. This study is a proof-of-concept experiment designed to test whether inhibition of IL-6 signalling leads to changes in psychotic symptoms. It could also inform likely statistical power for future trials testing efficacy of the drug as a treatment of schizophrenia, which is not the intention of this study. The exact statistical tests and techniques that will be applied to the data will depend on the objective of specific analysis and data characteristics (eg, variable type, distribution). These details will be specified in analysis plans and registered online before participants are unblinded and any data analysis is performed.

## Randomisation and blinding

An external agency independent of the study team will arrange random allocation to tocilizumab or normal saline group 1:1, ensuring two groups are comparable regarding anhedonia severity and sex. Randomisation will be stratified by site. Randomising agency will provide the randomisation code to the relevant hospital pharmacy who will dispense tocilizumab or normal saline according to the randomisation schedule. Dispensing pharmacies will keep a log of products dispensed. Infusions will be prepared and administered at clinical research facilities (CRFs). Infusion packs will be prepared by trained staff not part of the core study team, ensuring blinding of treatment allocation. Infusion packs containing drug or placebo will be visually indistinguishable from each other, ensuring that both participants and study team remain blind regarding treatment allocation.

## Statistical analysis

For randomised participants, an intention-to-treat approach will be taken for data analysis by including all randomised participants in statistical analyses, regardless of the treatment they received (if any). We will compare outcome measures between treatment and placebo groups controlling for baseline scores. This mechanistic experiment will focus on overall pattern of results and their effect sizes rather than p values for individual tests of statistical significance. The secondary mechanistic and observational analysis will compare psychotic symptoms, cognitive function, blood, neuroimaging and other biomarkers between and across study groups using appropriate statistical tests.

## STUDY PROCEDURE

An overview of study procedures is presented in figure 1 and all study measures are detailed in table 2. Recruitment will take place in Birmingham, Bristol and Cambridge and assessments at University and National Health Service (NHS) research facilities.

## Participant identification

Potential participants with psychosis will be identified by NHS Psychosis Early Intervention (EI) teams. HCs will be recruited through advertisement methods in Birmingham and Cambridge. Potential participants will complete a screening questionnaire to confirm their eligibility to participate. If deemed eligible, participants will be invited to an appointment to complete a full eligibility assessment.

## Eligibility assessment

Assessments will be carried out to establish eligibility and to obtain informed consent. Patients will complete the MINI to confirm ICD-10 diagnosis of schizophrenia and related psychoses, the PANSS to confirm the presence of positive symptoms of psychosis and the TEPS to confirm eligibility based on anticipatory and consummatory

pleasure sum scores. A blood sample will be collected from patients for serum IL-6 measurement. An MRI screening questionnaire will be administered to those willing to give informed consent for neuroimaging.

## Baseline assessment

All participants (60 inflamed psychosis, 30 non-inflamed psychosis and 30 HCs) will attend a baseline assessment comprising psychiatric measures, cognitive tasks, blood sampling and neuroimaging (optional). This will be the final study contact for patients without evidence of inflammation and HCs. Patients with evidence of inflammation will undergo further tests to establish safety/eligibility to receive tocilizumab, including an X-ray of the chest and blood tests to exclude pregnancy and certain infections, such as tuberculosis (TB), human immunodeficiency viruses (HIV), and COVID-19. Eligible participants will be randomised and invited for infusion.

## Intervention

Intravenous infusion of tocilizumab or normal saline will be given continuously over 1 hour at CRFs in Birmingham, Bristol and Cambridge by trained clinical staff under the supervision of a designated study doctor. Participants will remain under clinical observation for a further 1-hour period after the end of infusion.

## Follow-up assessments

Follow-up assessments will take place approximately 7, 14 and 28 days post infusion and will collect similar data to the baseline assessment. Cognitive tasks and neuroimaging (optional) will be administered only on day 14. Around 42 days post infusion, participants will be contacted by phone to provide a final debrief, at which point they will exit the study.

## RISK MANAGEMENT
## Psychosis-related risks

All patients will be under the care of a specialist NHS psychosis EI service. Participation will not involve any treatment modifications or significant delays in receiving treatment. If a patient becomes distressed during an assessment, or does not wish to continue for any reason, the researcher will stop the assessment. Participants may withdraw at any time without giving a reason. If there is any concern for the participant's safety, the research team will liaise with participant's general practitioner (GP) and/or mental health team as needed.

## Procedure-related risks
### Venepuncture

Blood taking is associated with mild discomfort and other side effects are rare. Efforts will be made to minimise discomfort. Blood taking will be performed by a nurse, doctor or research team member trained in venepuncture.

### X-ray of the chest

This study will use a typical effective radiation dose of 0.014 mSv; equivalent to 2.5 days of average natural

background radiation in the UK. The risk of developing cancer as a consequence of participating in this study is 0.0001%. Only non-pregnant, adult participants will be included.

### Neuroimaging

Discomfort during MRI will be minimised by using mirrors to allow participants to view outside of the machine, providing ear plugs and a panic button and allowing participants to communicate with the researcher and scan operator throughout. Mild transient vertigo may be experienced when being moved into the MRI machine. Risk of dislodgement or malfunction of medical implants or metallic foreign objects will be minimised by screening participants to ensure no metal is present on or within the body.

### IL-6 levels

We expect some 30%–50% of patient participants to have evidence of inflammation in the blood (IL-6≥0.7 pg/mL). This is not a cause for concern. Reasons for elevated IL-6 in the absence of an acute infection or chronic inflammatory illness could include obesity, smoking, alcohol use and lack of exercise, so knowledge of 'inflammation status' may prompt participants to adopt a healthier lifestyle. If the serum IL-6 level is high (ie, IL-6≥0.7 pg/mL) along with elevated CRP (>20 mg/L) without any apparent explanation, such as infection or chronic inflammatory illness, we will inform the participant's GP and the participant will be excluded from the study.

### Risk to research staff

Staff will follow local safety procedures when lone working. No other risks are anticipated.

### Safety considerations for infusion and monitoring of adverse reaction
#### Before infusion

Participants will be selected based on strict inclusion and exclusion criteria. Additionally, we will carry out tests for TB, HIV, varicella zoster virus (VZV) antibody and Hepatitis B and C because, though unlikely after a single dose, tocilizumab could make these infections worse. Female participants of childbearing age will be given a pregnancy test, which must be negative. Participants who are sexually active will be asked to use at least one form of effective contraception for 6 weeks post infusion. Male participants will also be asked not to donate sperm samples for 6 weeks post infusion.

#### During infusion

Infusions will be given under the supervision of a designated study doctor. Participants will be monitored for possible side effects, which will be managed in line with use of tocilizumab for treating patients with RA.

#### After infusion

Participants will remain under observation for 1 hour post infusion. Participants will be advised to seek help if they feel unwell after leaving the assessment centre and will be given an information sheet containing a telephone number their health professionals can call. If necessary, we will unblind the participant and inform their health professional whether they received tocilizumab or normal saline. Adverse reactions will be recorded at each follow-up visit. Additional, safety blood tests will be done at second follow-up (eg, white blood cell count, liver function, lipids).

## ETHICS AND DISSEMINATION

The study will be conducted in accordance with the Research Ethics Committee (REC), Health Research Authority (HRA) and local Research and Development department approvals and guidelines (REC reference: 22/EE/0010). The study team will prepare protocol amendments as required and ethics approval will be sought before implementing any changes to the approved protocol. The ISRCTN Trial Registry and the Research Governance Office will be informed of any amendments to the protocol.

### Consent

Informed consent will be obtained prior to eligibility assessments for participation in the study (online supplemental appendix I–III). This will include consent to randomise, for contact with their GP to inform them about participation, access GP/psychiatric records to verify medical history to establish eligibility and to inform the participant's GP any results/outcomes as necessary. Consent for additional tests to establish safety for tocilizumab infusion and for storing biological samples will also be obtained.

### Study management

The study is sponsored by the University of Bristol. The sponsor, the chief investigator (GMK) and the colead (RU) will have overall responsibility for the study. A named principal investigator will take clinical responsibility for study activities at each site. The study does not require the formal arrangement of a steering committee because, according to the HRA, it is not a Clinical Trial of an Investigational Medicinal Product. However, to enhance monitoring of the study, a study management group will be established, comprising academic and clinical experts in psychiatry, rheumatology, neuroscience and immunology.

### Data management and retention of samples

All potential participants will be assigned a unique study-specific participant ID number. All data will be subject to good practice as laid down in the Data Protection Act. Each study stage is tracked so that participant's (deidentified) status within the study is known, and assessment and other appointment dates are forecasted. This information is held in a secure, password-protected database. Anonymised data from assessments will be uploaded to a secure,

password-protected database using secure web-based data entry systems. Minimal personal data (age, sex) will be indexed by each participant's unique ID number. Blood samples collected in this study may be stored for up to 10 years after the completion of additional research. Stored samples will be coded throughout the sample storage and analysis process and will not be labelled with personal identifiers. Participants may withdraw their consent for their samples to be stored for future research.

## Dissemination plan

Study results will be published in peer-review journals and will conform to the guidelines of the International Committee of Medical Journal Editors. Findings will be disseminated at conferences, departmental talks and via social and traditional media.

## Author affiliations

[1] MRC Integrative Epidemiology Unit, Population Health Sciences, Bristol Medical School, University of Bristol, Bristol, UK
[2] Centre for Academic Mental Health, Population Health Sciences, Bristol Medical School, University of Bristol, Bristol, UK
[3] Institute for Mental Health and Centre for Human Brain Health, University of Birmingham, Birmingham, UK
[4] Department of Neuroscience and Behaviour, Division of Psychiatry, Ribeirão Preto Medical School, University of São Paulo, São Paulo, Brazil
[5] Institute of Clinical Sciences, College of Medical and Dental Sciences, University of Birmingham, Birmingham, UK
[6] Department of Psychiatry, University of Cambridge, Cambridge, UK
[7] Cambridgeshire and Peterborough NHS Foundation Trust, Fulbourn, UK
[8] Faculty of Biology, Medicine and Health, Division of Neuroscience and Experimental Psychology, School of Biological Sciences, University of Manchester, Manchester Academic Health Science Centre, Manchester, UK
[9] Department of Medicine, University of Cambridge, Cambridge, UK
[10] Early Intervention Service, Birmingham Women's and Children's NHS Foundation Trust, Birmingham, UK
[11] National Institute for Health and Care Research, Bristol Biomedical Research Centre, Bristol, UK
[12] Avon and Wiltshire Mental Health Partnership NHS Trust, Bristol, UK

**Twitter** Éimear M Foley @eimear_foley1

**Collaborators** The PIMS Collaboration: Golam Khandaker, Rachel Upthegrove, Alice Egerton, Anthony David, Bill Deakin, Carmine Pariante, David Cotter, Ed Bullmore, Eva Meisenzahl, Gary Donohoe, Georgios Gkoutos, Jack Rogers, James MacCabe, Joanna Neill, John Suckling, Neil Harrison, Nicholas Barnes, Nikos Koutsouleris, Paola Dazzan, Peter Jones, Stephen Burgess, Stephen Wood, Valeria Mondelli.

**Contributors** ÉMF wrote the first draft of the PIMS trial protocol and of this manuscript. SLG, MK, GMK, BD, DJ, JS, and NMB contributed to study design and protocol development and revised manuscript drafts. RU contributed to study design and study protocol and revised manuscript drafts. GMK devised study design and trial protocol and revised drafts. ÉMF and SLG developed study materials and liaised with REC and HRA regarding approvals. AM, JR, FC-Z, HH, EW, and MW contributed to the revision of the manuscript and validation of operating procedures and mechanistic protocols. RU and GMK co-lead the MRC grant that funds the PIMS trial and provided overall supervision and oversight for the project.

**Funding** The PIMS trial is funded by a Medical Research Council (MRC) grant to RU and GMK; Grant Ref: MR/S037675/1. ÉMF is supported by an MRC Integrative Epidemiology Unit PhD Studentship (MC_UU_00011/1). AM is supported by funding from the MRC for doctoral training (MR/2434208). FC-Z receives a PhD Fellowship from the São Paulo Research Foundation (2019/13229-2 and 2021/07448-3). HH is supported by the PIMS Trial MRC grant (MR/S037675/1). DJ is supported by the Cambridge Arthritis Research Endeavour and the National Institute for Health Research (NIHR) Cambridge Biomedical Research Centre (BRC-1215-20014). NMB acknowledges funding support from the MRC (MR/R006008/1 and MR/N019016/1), Ministry of Defence (702931454), Diabetes UK (20/0006296), NIHR (14/WM/0093) and Innovate UK (84361). RU has grants from MRC, NIHR: Health Technology Assessment, European Commission - Research: The Seventh Framework Programme and personal fees from Sunovion, outside the submitted work. RU is funded/supported by the NIHR Oxford Health Biomedical Research Centre. The views expressed are those of the author(s) and not necessarily those of the NIHR or the Department of Health and Social Care. GMK acknowledges funding support from the Wellcome Trust (grant number 201486/Z/16/Z), the MRC, UK (grant number MC_UU_00011/1; grant number MR/S037675/1; and grant number MR/W014416/1) and the NIHR Bristol Biomedical Research Centre, UK (grant number NIHR203315). The funders had no role in the design of this study.

**Competing interests** NMB holds shares and is a Director of Celentyx, other authors have no conflicts of interest to report.

**Patient and public involvement** Patients and/or the public were involved in the design, or conduct, or reporting, or dissemination plans of this research. Refer to the Methods section for further details.

**Patient consent for publication** Not applicable.

**Provenance and peer review** Not commissioned; externally peer reviewed.

**ORCID iDs**
Éimear M Foley http://orcid.org/0000-0002-3603-3774
Graham K Murray http://orcid.org/0000-0001-8296-1742

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
