## [Reviewer comments · BMJ Open]

ARTICLE DETAILS

TITLE (PROVISIONAL)	Protocol for the Psychosis Immune Mechanism Stratified Medicine (PIMS) trial: A randomised double-blind placebo-controlled trial of single dose tocilizumab in patients with psychosis
AUTHORS	Foley, Éimear; Griffiths, Sian Lowri; Murray, Alexander; Rogers, Jack; Corsi-Zuelli, Fabiana; Hickinbotham, Hannah; Warwick, Ella; Wilson, Martin; Kaser, Muzaffer; Murray, Graham K.; Deakin, Bill; Jadon, Deepak; Suckling, John; Barnes, Nicholas; Upthegrove, Rachel; Khandaker, Golam M.

VERSION 1 – REVIEW

REVIEWER	Hansen , Niels University of Gottingen Center of Internal Medicine
REVIEW RETURNED	18-Oct-2022

GENERAL COMMENTS	The authors present an excellent study protocol that improves knowledge of the important use of monoclonal antibodies such as tocilizumab in patients with psychosis. The protocol is very interesting and well written. I have some important questions that should be addressed in a major revision: The rationale is not clear why only anhedonia was used and not a broader psychopathology that addresses negative symptoms in psychosis. It would also be interesting to include additional positive symptoms, such as auditory hallucinations, in the study to better explore the hypothesis of poor functional outcomes, particularly in anhedonia and amotivation. The authors perform immunophenotyping in blood, which is excellent, but I recommend that activated CD4+ T cells and CD8+ T cells (HLADR+) also be examined. In addition, the authors should also decide whether the patient agrees to collect CSF for immunophenotyping with flow cytometry. The authors should discuss why they are only looking for peripheral inflammation and not CNS inflammation, which would be more useful, or at least address the limitations of looking for peripheral inflammation and the relationship to a central CNS process that might affect psychopathology.
---

REVIEWER	Singh, Harmanjit Government Medical College and Hospital Department of Pharmacology
REVIEW RETURNED	27-Oct-2022

GENERAL COMMENTS	1. Sample size must be calculated based on some specific assumptions. simply taking 60 patients is not the appropriate method. 2. Statistical tests and techniques to be used in this study should be specified 3. Control group must be defined more carefully and how the comparison would be done in the absence of the same pathogenesis in the control group 4. What is the rationale behind the cognitive assessments after 7, 14, 28 and telephonically after day 42, why not long term?
---

REVIEWER	Randell, Rachel Duke University School of Medicine
REVIEW RETURNED	27-Oct-2022

GENERAL COMMENTS	This is a well written and thorough protocol describing a randomized, proof-of-concept trial of tocilizumab (monoclonal antibody directed against the receptor for the pro-inflammatory cytokine IL-6) for psychosis in individuals with evidence of systemic inflammation. For the most part, the trial is well justified although there are a few gaps, described below. The greatest threat to this study is that endpoint assessment at 7, 14, and 28 days after a single dose of tocilizumab may fail to capture any meaningful effects on IL-6 signaling and/or symptoms (as the authors acknowledge on page 19, lines 271-274). However, this is a proof-of-concept trial designed to examine the causal role of IL-6 and obtain preliminary results to inform future efficacy studies. The proposed trial seems clinically meaningful and results will contribute both to the understanding of disease pathogenesis and planning of future therapeutic efficacy studies. Comment on the need to stratify or exclude based on obesity status (BMI \geq 30) Additional justification for the selection of anhedonia as primary outcome is needed Additional justification for threshold of serum IL-6 \geq 0.7pg/mL as evidence of inflammation is needed The investigators should consider collecting IL-6 samples at the same time of day, or at least recording in case the results require adjustment for diurnal variation in serum IL-6 levels (see Nilsson G, Lekander M, Åkerstedt T, et al. PLoS One. 2016 Nov 10;11(11):e0165799)
---

VERSION 1 – AUTHOR RESPONSE

Reviewer: 1

Dr. Niels Hansen, University of Gottingen Center of Internal Medicine

Comments to the Author:

The authors present an excellent study protocol that improves knowledge of the important use of monoclonal antibodies such as tocilizumab in patients with psychosis. The protocol is very interesting and well written. I have some important questions that should be addressed in a major revision:

Reply: We thank the reviewer for their kind words and effort in reviewing this manuscript. Our response to each of the reviewer's comments can be found below.

The rationale is not clear why only anhedonia was used and not a broader psychopathology that addresses negative symptoms in psychosis.

Reply: Thank you for allowing us to clarify this. Schizophrenia is a phenotypically heterogeneous syndrome with a diverse repertoire of symptoms. It is possible that not all features of the illness are related to inflammation. For instance, symptom-based studies of depression suggest that IL-6 and CRP are particularly associated with somatic (e.g., fatigue, anhedonia, appetite/sleep disturbance) rather than psychological (e.g., hopelessness, excessive/inappropriate guilt) symptoms [1–3]. Emerging evidence also indicates evidence for symptom specificity in psychosis. Meta-analytic data suggests that elevated proinflammatory cytokines are associated with negative psychotic symptomatology [4]. Moreover, a recent study from the ALSPAC birth cohort reported that out of 20 positive and negative symptoms, CRP is particularly associated with anhedonia and auditory hallucinations [5]. Lastly, results from work we have completed to date as part of the MRC-funded larger PIMS collaboration (MR/S037675/1), suggest that anhedonia may be a promising target in early phases of established psychotic disorder. Anhedonia and amotivation are strongly associated with poor functional outcomes in people with schizophrenia and often present a formidable barrier to returning to work or building relationships [6]. Focusing on particular inflammation-related symptoms may increase the chance of success for immunotherapy trials. As a result, the primary aim of this proof-of-concept trial is to test the effect of IL-6 inhibition on anhedonia in patients with psychosis.

Our trial's secondary objective is to examine characteristics of inflammation-associated psychosis. Table 2 details all study measures in the PIMS trial. This includes 1) the Positive and Negative Syndrome Scale (PANSS), which assesses the presence and severity of positive and negative symptoms, as well as general psychopathology in patients with psychosis, and 2) The Scale for the Assessment of Negative Symptoms (SANS), which evaluates five different aspects of negative symptoms: alogia, affective blunting, avolition-apathy, anhedonia-asociality, and attentional impairment. Therefore, while the PIMS trial is focused primarily on anhedonia as an outcome, it also makes use of several validated scales to assess a broader psychopathology, including an array of both positive and negative symptoms.

This rationale has been further clarified in the Introduction section of the manuscript (see pages 8 and 9).

It would also be interesting to include additional positive symptoms, such as auditory hallucinations, in the study to better explore the hypothesis of poor functional outcomes, particularly in anhedonia and amotivation.

Reply: Thank you for this suggestion. As detailed above, the PIMS Trial is comprised of several psychiatric measures recorded at several timepoints across the study (see Table 2 for details). This includes the PANSS which contains a scale dedicated to the assessment of positive symptoms, including hallucinatory behaviour (auditory, visual, olfactory, or somatic realms). Unfortunately, as the trial has now received ethical approval, no further changes can be made to the protocol.

The authors perform immunophenotyping in blood, which is excellent, but I recommend that activated CD4+ T cells and CD8+ T cells (HLADR+) also be examined. In addition, the authors should also decide whether the patient agrees to collect CSF for immunophenotyping with flow cytometry.

Reply: We thank the reviewer for this suggestion. We agree that activated T cells are important to examine in this population. Flow cytometry will be conducted on peripheral blood samples, rather than CSF to minimise patient burden, and a range of T cell subsets will be examined, including CD4+ T cells and CD8+ T cells. As the trial has now received ethical approval, no further changes can be made to the protocol to include CSF as a measure.

The authors should discuss why they are only looking for peripheral inflammation and not CNS inflammation, which would be more useful, or at least address the limitations of looking for peripheral inflammation and the relationship to a central CNS process that might affect psychopathology.

Reply: CSF sampling, in addition to the extensive blood and neuroimaging work was felt to be too burdensome for this patient population. Unlike some EU countries, in the UK it is not usual to have routine lumbar puncture performed in patients with first-episode psychosis. Moreover, the PIMS Trial involves the collection of neuroimaging and cognitive function data that will give an insight into brain relevance.

Reviewer: 2

Dr. Harmanjit Singh, Government Medical College and Hospital Department of Pharmacology

Comments to the Author:

1. Sample size must be calculated based on some specific assumptions. simply taking 60 patients is not the appropriate method.

Reply: We agree with the reviewer that a sample size calculation would be useful for this study. However, currently there are no trials of immunotherapies for anhedonia in schizophrenia, making accurate power calculation difficult. Moreover, as this is an experimental medicine proof-of-concept trial, our study does not need to be sufficiently powered to detect a clinically meaningful outcome. We hope that this study can inform likely statistical power/sample size for future trials testing efficacy of the drug as a treatment of schizophrenia. Testing efficacy of the drug as a treatment for schizophrenia and related psychoses is not the intention for this study. This information is detailed in the "Sample size and statistical power" section of the manuscript (see page 17).

2. Statistical tests and techniques to be used in this study should be specified

Reply: For randomised participants (trial cohort), an intention-to-treat approach will be taken for data analysis. We will compare outcome measures between treatment and placebo groups controlling for baseline scores. This mechanistic experiment will focus on overall pattern of results rather than *P*-values for individual tests of statistical significance. The secondary, observational analysis will compare psychotic symptoms, cognitive function, blood, and other biomarkers between and across study groups using appropriate statistical tests. This information is detailed in the "Statistical Analysis" section of the manuscript.

The exact statistical tests and techniques that will be applied to the data will depend on the objective of specific analysis and data characteristics (e.g., variable type, distribution). These details will be specified in analysis plans and registered online before participants are unblinded and any data analysis is performed. This information has now been clarified in the manuscript (see pages 17 and 18).

3. Control group must be defined more carefully and how the comparison would be done in the absence of the same pathogenesis in the control group

Reply: Control participants are required to meet identical inclusion criteria to patients with psychosis, barring criteria related to psychiatric diagnosis, for which they must have no current or lifetime history. Control participants displaying any of the detailed exclusion criteria (see Table 1) will be excluded. These exclusion criteria are also identical for all control and patient participants. Having a control group will allow us to first demonstrate any illness specific markers of pathogenesis (i.e., control vs

schizophrenia), and then second stage of immune active vs non-immune active participants. Control data is also essential for imaging data in some of the planned clustering analyses.

4. What is the rationale behind the cognitive assessments after 7, 14, 28 and telephonically after day 42, why not long term?

Reply: Thank you for allowing us to clarify this. Cognitive assessments will be performed at the baseline assessment and day 14 post-infusion only. These assessments will not take place telephonically, but rather face-to-face. Approximately 42 days post-infusion, participants will be contacted by phone to provide a final debrief, at which point they will exit the study.

Our trial involves a single intravenous infusion of tocilizumab, a humanized monoclonal antibody against the IL-6R licensed in the UK for treatment of rheumatoid arthritis. A single tocilizumab infusion has been shown to improve clinical and laboratory measures, including CRP, within 48 hours, with most noticeable results in one-to-two weeks [7,8]. In clinical practice, tocilizumab is given at a monthly dose (without a loading dose to begin with). Our follow-up schedule reflects these timings. Therefore, our primary follow-up assessment takes place two weeks after infusion, along with additional follow-ups at days 7 and 28. Finally, participants will be debriefed and exit the study 42 days post-infusion, because of the high likelihood this single dose of tocilizumab being clear of the system at this time point [9]. The PIMS trial is not conducting longer term dosing as it is a proof-of-concept trial (not an efficacy trial) designed to examine potential mechanisms by which IL-6 affects cognitive and clinical outcomes in psychosis, and our aim is to obtain preliminary results to inform future efficacy trials.

Reviewer: 3

Dr. Rachel Randell, Duke University School of Medicine

Comments to the Author:

This is a well written and thorough protocol describing a randomized, proof-of-concept trial of tocilizumab (monoclonal antibody directed against the receptor for the pro-inflammatory cytokine IL-6) for psychosis in individuals with evidence of systemic inflammation. For the most part, the trial is well justified although there are a few gaps, described below. The greatest threat to this study is that endpoint assessment at 7, 14, and 28 days after a single dose of tocilizumab may fail to capture any meaningful effects on IL-6 signaling and/or symptoms (as the authors acknowledge on page 19, lines 271-274). However, this is a proof-of-concept trial designed to examine the causal role of IL-6 and obtain preliminary results to inform future efficacy studies. The proposed trial seems clinically meaningful and results will contribute both to the understanding of disease pathogenesis and planning of future therapeutic efficacy studies.

Reply: Thank you for your kind comments and for your time and effort in reviewing this work. We agree with the reviewer that as a proof-of-concept study, this trial is limited in its ability to test the efficacy of the drug as a treatment of schizophrenia. However, as the reviewer has correctly highlighted, this is not the intention of our proof-of-concept study. We aim to test whether inhibition of IL-6 signalling leads to changes in psychotic symptoms and hope to inform likely statistical power for future efficacy trials. We know from previous studies that a single tocilizumab infusion can improve clinical and laboratory measures, including CRP, within 48 hours, with most noticeable results in one-to-two weeks [7,8]. In clinical practice, tocilizumab is given at a monthly dose (without a loading dose to begin with). Our follow-up schedule reflects these timings. Therefore, our primary follow-up assessment takes place two weeks after infusion, along with additional follow-ups at days 7 and 28. Finally, participants will be debriefed and exit the study 42 days post-infusion, because of the high likelihood of tocilizumab being clear of the system at this time point [9]. Our aim is to examine whether

change in immune markers mirror changes in clinical and cognitive measures in the short term and to obtain preliminary results to inform future efficacy trials.

Comment on the need to stratify or exclude based on obesity status (BMI \geq 30)

Reply: Participants with BMI >35 will be excluded from the trial due to this group being deemed at higher risk of COVID-19 complications. This criterion is in place to minimise risk to patients entering the trial during the COVID-19 outbreak.

Additional justification for the selection of anhedonia as primary outcome is needed

Reply: Thank you for allowing us to clarify this. Schizophrenia is a phenotypically heterogeneous syndrome with a diverse repertoire of symptoms. It is possible that not all features of the illness are related to inflammation. For instance, symptom-based studies of depression suggest that IL-6 and CRP are particularly associated with somatic (e.g., fatigue, anhedonia, appetite/sleep disturbance) rather than psychological (e.g., hopelessness, excessive/inappropriate guilt) symptoms [1–3]. Emerging evidence also indicates evidence for symptom specificity in psychosis. Meta-analytic data suggests that elevated proinflammatory cytokines are associated with negative psychotic symptomatology [4]. Moreover, a recent study from the ALSPAC birth cohort reported that out of 20 positive and negative symptoms, CRP is particularly associated with anhedonia and auditory hallucinations [5]. Lastly, results from work we have completed to date as part of the MRC-funded larger PIMS collaboration (MR/S037675/1), suggest that anhedonia may be a promising target in early phases of established psychotic disorder. Anhedonia and amotivation are strongly associated with poor functional outcomes in people with schizophrenia and often present a formidable barrier to returning to work or building relationships [6]. Focusing on particular inflammation-related symptoms may increase the chance of success for immunotherapy trials. As a result, the primary aim of this proof-of-concept trial is to test the effect of IL-6 inhibition on anhedonia in patients with psychosis.

Our trial's secondary objective is to examine characteristics of inflammation-associated psychosis. Table 2 details all study measures in the PIMS trial. This includes 1) the Positive and Negative Syndrome Scale (PANSS), which assesses the presence and severity of positive and negative symptoms, as well as general psychopathology in patients with psychosis, and 2) The Scale for the Assessment of Negative Symptoms (SANS), which evaluates five different aspects of negative symptoms: alogia, affective blunting, avolition-apathy, anhedonia-asociality, and attentional impairment. Therefore, while the PIMS trial is focused primarily on anhedonia as an outcome, it also makes use of several validated scales to assess a broader psychopathology, including an array of both positive and negative symptoms.

This rationale has been further clarified in the Introduction section of the manuscript (see pages 8 and 9).

Additional justification for threshold of serum IL-6 \geq 0.7pg/mL as evidence of inflammation is needed.

Reply: The threshold of serum IL-6 \geq 0.7pg/mL as evidence of inflammation for this particular trial was chosen based on observations from the Personalised Prognostic Tools for Early Psychosis Management (PRONIA) cohort [<https://www.pronia.eu>]. In 192 first-episode psychosis patients included in the PRONIA study, the median value of serum IL-6 was 0.49pg/mL (25th percentile 0.22pg/mL; 75th percentile 1.11pg/mL), and the mean was 0.79pg/mL (SD \pm 0.84). Based on these

observations, we chose the cut-off of 0.7pg/mL for patient selection in the PIMS trial. This information has now been added to the manuscript (see page 12).

The investigators should consider collecting IL-6 samples at the same time of day, or at least recording in case the results require adjustment for diurnal variation in serum IL-6 levels (see Nilsson G, Lekander M, Åkerstedt T, et al. PLoS One. 2016 Nov 10;11(11):e0165799)

Reply: Thank you for this suggestion. We envisage blood sampling will take place during working hours, but we have not specified a time to ease burden on patients and to maximise participation. However, we will record time of blood sampling so we can take into account any effect of diurnal variation. This information has been added to the manuscript (see page 15).

Reviewer: 1

Competing interests of Reviewer: I have no competing interests.

Reviewer: 2

Competing interests of Reviewer: None

Reviewer: 3

Competing interests of Reviewer: Dr. Randell is supported by the Eunice Kennedy Shriver National Institute of Child Health & Human Development of the NIH under Award Number T32HD104576. Dr. Randell's spouse has financial relationships with Merck & Co, and Biogen.

References

- 1 Chu AL, Stochl J, Lewis G, *et al.* Longitudinal association between inflammatory markers and specific symptoms of depression in a prospective birth cohort. *Brain Behav Immun* 2019;**76**:74–81. doi:10.1016/j.bbi.2018.11.007
- 2 Jokela M, Virtanen M, Batty GD, *et al.* Inflammation and Specific Symptoms of Depression. *JAMA Psychiatry* 2016;**73**:87–8. doi:10.1001/jamapsychiatry.2015.1977
- 3 Lamers F, Milaneschi Y, de Jonge P, *et al.* Metabolic and inflammatory markers: associations with individual depressive symptoms. *Psychol Med* 2018;**48**:1102–10. doi:10.1017/S0033291717002483
- 4 Dunleavy C, Elsworth RJ, Uptegrove R, *et al.* Inflammation in first-episode psychosis: The contribution of inflammatory biomarkers to the emergence of negative symptoms, a systematic review and meta-analysis. *Acta Psychiatrica Scandinavica* 2022;**146**:6–20. doi:10.1111/acps.13416
- 5 Khandaker GM, Stochl J, Zammit S, *et al.* Association between circulating levels of C-reactive protein and positive and negative symptoms of psychosis in adolescents in a general population birth cohort. *J Psychiatr Res* 2021;**143**:534–42. doi:10.1016/j.jpsychires.2020.11.028
- 6 Edwards CJ, Cella M, Tarrier N, *et al.* Investigating the empirical support for therapeutic targets proposed by the temporal experience of pleasure model in schizophrenia: A systematic review. *Schizophr Res* 2015;**168**:120–44. doi:10.1016/j.schres.2015.08.013
- 7 Choy EHS, Isenberg DA, Garrood T, *et al.* Therapeutic benefit of blocking interleukin-6 activity with an anti-interleukin-6 receptor monoclonal antibody in rheumatoid arthritis: a randomized, double-blind, placebo-controlled, dose-escalation trial. *Arthritis Rheum* 2002;**46**:3143–50. doi:10.1002/art.10623
- 8 Woo P, Wilkinson N, Prieur A-M, *et al.* Open label phase II trial of single, ascending doses of MRA in Caucasian children with severe systemic juvenile idiopathic arthritis: Proof of principle of the efficacy of IL-6 receptor blockade in this type of arthritis and demonstration of prolonged clinical improvement. *Arthritis research & therapy* 2005;**7**:R1281-8. doi:10.1186/ar1826

9 Woo P, Wilkinson N, Prieur A-M, *et al.* Open label phase II trial of single, ascending doses of MRA in Caucasian children with severe systemic juvenile idiopathic arthritis: proof of principle of the efficacy of IL-6 receptor blockade in this type of arthritis and demonstration of prolonged clinical improvement. *Arthritis Res Ther* 2005;7:R1281–8. doi:10.1186/ar1826

VERSION 2 – REVIEW

REVIEWER	Hansen , Niels University of Gottingen Center of Internal Medicine
REVIEW RETURNED	13-Dec-2022

GENERAL COMMENTS	The manuscript has substantially improved. It can be endorsed for publication.
--

REVIEWER	Randell, Rachel Duke University School of Medicine
REVIEW RETURNED	20-Dec-2022

GENERAL COMMENTS	I thank the authors for thoroughly addressing our comments and making appropriate modifications to the manuscript. As the trial has already begun enrolling patients, it is understandable that no changes to the protocol were made. I look forward to future results of the PIMS trial.
---

VERSION 2 – AUTHOR RESPONSE

Reviewer: 1

Dr. Niels Hansen, University of Gottingen Center of Internal Medicine

Comments to the Author:

The manuscript has substantially improved. It can be endorsed for publication.

Reply: Thank you.

Reviewer: 3

Dr. Rachel Randell, Duke University School of Medicine

Comments to the Author:

I thank the authors for thoroughly addressing our comments and making appropriate modifications to the manuscript. As the trial has already begun enrolling patients, it is understandable that no changes to the protocol were made. I look forward to future results of the PIMS trial.

Reply: Thank you.

Reviewer: 1

Competing interests of Reviewer: No competing interests.

Reviewer: 3

Competing interests of Reviewer: Dr. Randell is supported by the Eunice Kennedy Shriver National Institute of Child Health & Human Development of the NIH under Award Number T32HD104576. Dr. Randell's spouse has financial relationships with Merck & Co, and Biogen.